Examination of optical coherence tomography findings in patients with pregabalin use disorder

Kılıç Osman Hasan Tahsin 1 hasan.kilic@idu.edu.tr
Bayram Zehra Nur 2
Kiyat Pelin 3
Karti Omer 3
Aral Arzu 2
Munis Nazlı Deniz 2
Mutlu Berfin Gurbet 2
1 Department of Psychiatry, Faculty of Medicine, İzmir Democracy University , İzmir , Turkey
2 Department of Psychiatry, Institute of Health Sciences, İzmir Democracy University , İzmir , Turkey
3 Department of Ophthalmology, Buca Seyfi Demirsoy Training and Research Hospital, İzmir Democracy University , İzmir , Turkey
Fogt Nick
Electronic publication date: 2024 Nov 11
Publication date: 2024
Volume: 12
Electronic Location ID: e18395
Received 2024 Apr 10; Accepted 2024 Oct 3
Copyright: © 2024 Kılıç et al.
Copyright year: 2024
Copyright holder: Kılıç et al.
License: This is an open access article distributed under the terms of the Creative Commons Attribution License, which permits unrestricted use, distribution, reproduction and adaptation in any medium and for any purpose provided that it is properly attributed. For attribution, the original author(s), title, publication source (PeerJ) and either DOI or URL of the article must be cited.
License URL: https://creativecommons.org/licenses/by/4.0/

Keywords: Pregabalin use disorder, Optic coherence tomography, Retinal nerve fiber layer, Ganglion cell complex

Funding: This research did not receive any specific grant from funding agencies in the public, commercial, or not-for-profit sectors.

==============================
Background

Pregabalin abuse is a rapidly growing health problem worldwide, and little is known about the effects of prolonged high-dose use in patients with pregabalin use disorder.

Objective

In this study, the effects of pregabalin abuse on retinal layers were investigated in patients with pregabalin use disorder (PGUD).

Methods

This study included 35 controls and 34 patients with PGUD, according to Diagnostic and Statistical Manual of Mental Disorders (DSM)-5 criteria. Optic coherence tomography (OCT) measurements including the retinal nerve fiber layer (RNFL), ganglion cell layer-inner plexiform layer (GCL-IPL) and ganglion cell complex (GCC) were performed. RNFL thickness was evaluated in four quadrants (inferior, superior, nasal, temporal). GCL-IPL and GCC thickness were evaluated in six sectors (superior, superonasal, inferonasal, inferior, inferotemporal, superotemporal).

Results

GCC inferonasal (p = 0.040, r = 0.354), GCC inferior (p = 0.018, r = 0.402) GCL-IPL inferior (p = 0.031, r = 0.370) and GCL-IPL inferotemporal (p = 0.029, r = 0.376) thickness were positively correlated with the duration of pregabalin use. There was no significant sector or quadrant-wise difference between groups (p > 0.05).

Conclusion

Our findings emphasized the drug’s potential neuroprotective effect. It should be taken into consideration that neurodegenerative changes due to substance use disorder occur with long-term. Longitudinal prospective studies investigating dose-duration relationship are needed.

Introduction

Pregabalin abuse is a rapidly growing health problem worldwide (Evoy et al., 2021). Although early publications reported high levels of pregabalin abuse among those with opioid use disorder, recent studies have shown that its prevalence varies between 0.5% and 8.5% in the general population and is at a significant level even among those without a history of substance use (Hägg, Jönsson & Ahlner, 2020; Baldwin & Masdrakis, 2023). Additionally, self-harm, criminal behavior, traffic accidents, and injuries have increased among pregabalin abusers (Abrahamsson et al., 2017; Molero et al., 2019).

Pregabalin is a γ-aminobutyric acid (GABA) analogue that indirectly reduces neuronal excitability and firing through various molecular interactions (Bockbrader et al., 2010). At therapeutic doses, the most reported side effects are dizziness, sleepiness, dry mouth, edema, and blurred vision (Baldwin et al., 2015). Moreover, pregabalin exhibits a dose-dependent euphoric effect, which is considered responsible for its abuse potential (Zaccara et al., 2017). Although the majority of pregabalin abusers do not experience serious toxicity, complications such as seizures, bradycardia, hypotension, pneumonia, coma, and death have been reported (Antunovic et al., 2023). However, little is known about the effects of long-term high-dose use.

Long-term and high-dose use of addictive substances has been shown to have neurotoxic and neurodegenerative effects. Through microbiota changes and increased oxidative stress, substances like alcohol and methamphetamine can lead to neuroinflammation and consequently neurodegeneration (Hillemacher et al., 2018; He et al., 2023; Kwon & Koh, 2020). Neurodegenerative effects of cocaine have also been reported, with mitochondrial dysfunction considered the main pathophysiological mechanism (Guo, Chen & Wang, 2023). Similarly, the potential neurotoxic and neurodegenerative effects of pregabalin have been demonstrated in animal studies (Kamel, 2016; Elsukary et al., 2022). Moreover, an Optical Coherence Tomography (OCT) study examining the long-term use of pregabalin at therapeutic doses reported significant thinning of the retinal nerve fiber layer, indicating potential neurodegeneration (Biçer et al., 2022).

Optical Coherence Tomography is a non-invasive, inexpensive, rapid, and radiation-free method that measures the layers of the retina and optic nerve, which are considered direct extensions of the brain (Fujimoto, 2003). The retinal layers measured with the OCT device include the ganglion cell layer (GCL), inner plexiform layer (IPL), and retinal nerve fiber layer (RNFL). These three layers form the ganglion cell complex (GCC). GCL consists of ganglion cell bodies, IPL comprises retinal ganglion cell dendrites, and RNFL represents ganglion cell axons (Khalil & Said, 2019). Thinning of these layers can be considered an objective and sensitive parameter of neurodegeneration in neurological and psychiatric disorders (Orum & Kalenderoglu, 2020; Xiao et al., 2023).

However, the number of studies evaluating OCT measurements in substance use disorders is limited. Thinning of the retinal layers has been detected in individuals with alcohol and cocaine use disorders (Gemelli et al., 2019; Orum & Kalenderoglu, 2020). Currently, there is no study evaluating OCT parameters in individuals with pregabalin use disorder. Therefore, this study aims to compare RNFL, GCL-IPL (between the inner limiting membrane and the inner nuclear layer), and GCC thickness in patients with pregabalin use disorder with those of age- and sex-matched healthy volunteers.

Materials and Methods

The Ethics Committee of İzmir Democracy University Buca Seyfi Demirsoy Training and Research Hospital in İzmir, Türkiye, approved the research protocol (approval number 2023/178). All participants were informed of the study’s purpose and provided written informed consent. The study was conducted in strict accordance with the principles of the Declaration of Helsinki.

Subjects and clinical assessments

Between December 2023 and February 2024, 467 individuals applied to the probation clinic at İzmir Democracy University. In accordance with Turkey’s national policy, individuals involved in the use, purchase, or possession of controlled substances are mandated to attend the probation clinic. All participants were administered the SCID-5-CV (Structured Clinical Interview for DSM-5) (First et al., 2016) by the same psychiatrist. Comprehensive ophthalmologic examinations were conducted in the Department of Ophthalmology, including assessments of best corrected visual acuity (BCVA), intraocular pressure, slit-lamp biomicroscopy, and fundus examination. The control group comprised hospital employees, including secretaries, cleaning staff, and security personnel, who had no history of substance use other than tobacco and no systemic medical conditions or medication usage.

Inclusion/exclusion criteria

Male participants aged between 18 and 45 were included in the study. Participants with comorbid psychiatric disorders, except for tobacco use disorder and personality disorders, were excluded. Exclusions also applied to individuals with neurological, immunological, or systemic diseases, as well as primary ophthalmological conditions such as glaucoma or retinal diseases. Additionally, participants with conditions that could affect image quality or measurement values, including high refractive errors and cataracts, were excluded. Only patients and controls with normal eye findings—defined as an intraocular pressure (IOP) lower than 21 mmHg, a refraction lower than 0.50 diopters both spherically and cylindrically, and no anterior or posterior segment pathologies as detected by slit-lamp biomicroscopy—were included in the study. Of the 467 individuals who applied to the probation clinic, 102 were diagnosed with Pregabalin Use Disorder (PGUD) following psychiatric evaluations and toxicological analyses. Of these, 12 were excluded due to being outside the age range of 18 to 45 years, and two females were excluded from the study. After excluding individuals with primary ophthalmological diseases, with conditions that could affect image quality or measurement values, those taking medications, and polysubstance users, the final analysis included 35 controls, and 34 participants diagnosed with Pregabalin Use Disorder (PGUD).

Spectral domain optical coherence tomography measurements

Spectral Domain Optical Coherence Tomography (SD-OCT) (DRI-OCT Triton, Topcon, Inc., Tokyo, Japan) measurements were performed to analyze RNFL, GCL-IPL, and GCC thickness. The images were evaluated blindly by two ophthalmologists in the Department of Ophthalmology. RNFL thickness values were recorded across four quadrants (superior, temporal, inferior, and nasal). GCL-IPL and GCC thickness values were measured in six sectors (superior, superotemporal, superonasal, inferior, inferotemporal, and inferonasal) (Figs. 1 and 2). The SD-OCT device’s automatic calculation and reporting system was used to analyze these values within each group. OCT measurements were taken for both eyes, but only the data from the left eye were used for statistical analysis.

Figure 1 Spectral domain optical coherent tomography image showing GCC measurement values in six sectors.

Figure 2 Spectral domain optical coherent tomography image showing RNLF measurement values in four quadrants.

Statistical methods

Statistical analyses were conducted using SPSS 25 (IBM Corporation, Armonk, New York, USA). Descriptive statistics were calculated as mean, frequency and percentage. Prior to comparisons, all data distributions were assessed using the Kolmogorov-Smirnov test. Categorical data were compared using the Pearson Chi-square test. Parametric tests (Independent t-test) were employed for normally distributed data, while non-parametric tests (Mann-Whitney U) were utilized for non-normally distributed data. Pearson correlation analysis was used to examine the relationships between the duration of Pregabalin use, age, and OCT parameters.

All retinal layers were compared between the control group and the Pregabalin group, and regression analyses were conducted to understand the effect of duration and smoking on the layers. Linear regression analyses were conducted to evaluate the potential effect of usage duration (in months) on different OCT measurements. A general linear model (GLM) was used to assess the impact of smoking and pregabalin use on all OCT measurements between the pregabalin and the control group. To evaluate the interactions between groups and the potential effects of smoking, group (PGUD/Control), smoking status, and the interaction between these two factors were included in the model. A p-value of < 0.05 was considered statistically significant for all tests.

Results

Socio-demographic and clinical data

The study included thirty-four patients with PGUD and thirty-five controls. The mean age of the PGUD and control groups was 30.67 ± 7.59 and 30.71 ± 9.29, respectively. There was no significant difference between the groups in terms of age (p = 0.000). All participants in both groups were male. The proportion of smokers was higher in the PGUD group (p = 0.000). Thirty-two (94.1%) of the patients with PGUD and twelve (34.2%) of the controls were smokers. There was no significant difference between the groups in terms of marital status (p = 0.184). In the PGUD group, 14 participants (41.1%) completed primary education, 14 (41.1%) had secondary education, and six (17.6%) completed high school. In the control group, eight participants (22.8%) completed secondary education, eight (22.8%) completed high school, and 19 (54.2%) had undergraduate education. The difference in educational attainment between the groups was statistically significant (p < 0.001). In the PGUD group, 21 participants (61.8%) were employed, while 13 participants (38.2%) were unemployed. Additionally, 16 participants (47.1%) in the PGUD group had health insurance, whereas 18 participants (52.9%) did not. In the control group, all participants were hospital staff, meaning they were both employed and had health insurance (100%). Therefore, no comparisons were made between the groups regarding employment status and health insurance coverage (not applicable (N/A)). The mean duration of pregabalin use in the PGUD group was 62.26 ± 40.09 months (min: 12, max: 153), and the mean daily dosage was 1,129.41 ± 601.40 mg (min: 300, max: 3,600). The socio-demographic and clinical features of the groups are given in Table 1.

Table 1 The sociodemographic characteristics of the participants.

	PGUD (34)	Controls (35)	p	
n (%)	n (%)	
Age (mean ± SD)	30.67 ± 7.59	30.71 ± 9.29	0.635a	
Marital status		13 (37.1%)	0.184b	
Married	13 (38.2%)	22 (62.9%)	
Single	18 (52.9%)		
Divorced	3 (8.8%)		
Smoking			0.000b	
Yes	32 (94.11%)	12 (34.2%)	
No	2 (5.9%)	23 (65.8%)	
Education	14 (41.1%)	0	0.000b	
Primary education	14 (41.1%)	8 (22.8%)	
Secondary education	6 (17.6%)	8 (22.8%)	
High school	0	19 (54.2%)	
Undergraduate education			
Postgraduate education			
Employment	21 (61.8%)	35 (100%)		
Yes	13 (38.2%)	0 (0%%)	
No			
Health insurance	16 (47.1%)	35 (100%)		
Yes	18 (52.9%)	0 (0%%)	
No			
Dosage (daily) (mean ± SD)	1,129.41 ± 7.59	Not applicable (N/A)		
Duration of use (month) (mean ± SD)	62.26 ± 40.09	Not applicable (N/A)		
Notes:

PGUD, pregabalin use disorder.

a Mann-Whitney U test,

b Chi-square test.

OCT findings

RNFL thickness was evaluated in four quadrants (inferior, superior, nasal, temporal). GCL-IPL and GCC thickness were evaluated in six sectors (superior, superonasal, inferonasal, inferior, inferotemporal, superotemporal). There was no significant difference between the groups in any sector or quadrant (p > 0.05 for all comparisons). OCT measurements are shown in Table 2. The impact of smoking and pregabalin use on all OCT measurements was assessed using General Linear Models (GLM). Overall, the models did not significantly explain the variance in any OCT measurements. For the RNFL measurements, none of the variables showed significant effects: RNFL-inferior (R2 = 0.003, F = 0.024, p = 0.878), RNFL-superior (R2 = 0.010, F = 0.001, p = 0.980), RNFL-nasal (R2 = 0.133, F = 0.987, p = 0.324), and RNFL-temporal (R2 = 0.047, F = 0.332, p = 0.566). Similarly, GCL-IPL measurements did not reveal any significant impact, including GCL-IPL superior (R2 = 0.033, F = 0.133, p = 0.717), GCL-IPL superonasal (R2 = 0.058, F = 0.026, p = 0.874), GCL-IPL inferonasal (R2 = 0.034, F = 0.236, p = 0.629), GCL-IPL inferior (R2 = 0.061, F = 0.901, p = 0.346), GCL-IPL inferotemporal (R2 = 0.017, F = 0.097, p = 0.756), and GCL-IPL superotemporal (R2 = 0.008, F = 0.046, p = 0.831). The GCC measurements were also unaffected, as indicated by GCC superior (R2 = 0.004, F = 0.195, p = 0.661), GCC superonasal (R2 = 0.021, F = 0.527, p = 0.470), GCC inferonasal (R2 = 0.004, F = 0.117, p = 0.734), GCC inferior (R2 = 0.011, F = 0.649, p = 0.424), GCC inferotemporal (R2 = 0.029, F = 0.007, p = 0.933), and GCC superotemporal (R2 = 0.007, F = 0.188, p = 0.666).

Table 2 Comparison of the OCT parameters.

	PGUD	Controls	p	
(mean ± SD)	(mean ± SD)		
RNLF-inferior	140.08 ± 18.28 μm	139.22 ± 18.71 μm	0.848a	
RNLF-superior	135.14 ± 18.33 μm	136.22 ± 15.02 μm	0.885b	
RNLF-nasal	82.82 ± 10.18 μm	88.37 ± 23.0 μm	0.601b	
RNLF-temporal	75.97 ± 10.07 μm	78.4 ± 9.67 μm	0.311a	
GCL-IPL superior	74.85 ± 5.50 μm	73.74 ± 6.38 μm	0.220b	
GCL-IPL superonasal	79.41 ± 4.78 μm	76.97 ± 6.11 μm	0.070a	
GCL-IPL inferonasal	77.17 ± 6.96 μm	75.71 ± 5.78 μm	0.346a	
GCL-IPL inferior	71.76 ± 6.88 μm	68.85 ± 12.52 μm	0.159b	
GCL-IPL inferotemporal	76.97 ± 6.12 μm	75.68 ± 6.74 μm	0.411a	
GCL-IPL superotemporal	74.50 ± 8.72 μm	74.91 ± 6.86 μm	0.407b	
GCC superior	111.8 ± 8.25 μm	112.31 ± 9.28 μm	0.749a	
GCC superonasal	124.52 ± 8.66 μm	122.94 ± 10.24 μm	0.490a	
GCC inferonasal	124.47 ± 9.35 μm	124.2 ± 10.08 μm	0.908a	
GCC inferior	110.14 ± 8.71 μm	110.4 ± 9.50 μm	0.670b	
GCC inferotemporal	106.73 ± 19.89 μm	101.85 ± 7.87 μm	0.238a	
GCC superotemporal	98.29 ± 11.94 μm	99.25 ± 7.78 μm	0.400b	
Notes:

a Independent t-test.

b Mann-Whitney U test.

PGUD, pregabalin use disorder; RNLF, retinal nerve fiber; GCL-IPL, ganglion cell layer inner plexiform layer; GCC, ganglion cell complex.

Correlations between pregabalin use and OCT parameters

RNFL superior (p = 0.011, r = −0.431), GCL-IPL superior (p = 0.011, r = −0.431), GCL-IPL superotemporal (p = 0.024, r = −0.386), GCL-IPL superonasal (p = 0.020, r = −0.300), and GCC inferonasal (p = 0.044, r = −0.348) measurements were negatively correlated with age. GCL-IPL inferior (p = 0.031, r = 0.370), GCL-IPL inferotemporal (p = 0.029, r = 0.376), GCC inferonasal (p = 0.040, r = 0.354), and GCC inferior (p = 0.018, r = 0.402) thickness were positively correlated with the duration of pregabalin use. Linear regression analyses were conducted to assess the impact of the duration of use (in months) on four retinal layers. The models for GCL-IPL inferior (R2 = 0.139, F (1,32) = 5.148, p = 0.030), GCC inferior (R2 = 0.147, F (1,32) = 5.521, p = 0.025), and GCL-IPL inferotemporal (R2 = 0.140, F (1,32) = 5.219, p = 0.029) were statistically significant, explaining 13.9%, 14.7%, and 14.0% of the variance, respectively. However, the model for GCC inferonasal did not reach statistical significance (R2 = 0.106, F (1,32) = 3.806, p = 0.060). The relationship between the duration of pregabalin use and the four retinal layers (GCL-IPL inferotemporal, GCL-IPL inferior, GCC inferonasal, and GCC inferior) is illustrated in Fig. 3. There was no other significant correlation between OCT measurements and dosage, age, or duration of use (p > 0.05). Dosage and all OCT measurements were negatively correlated, but none of them were at a significant level (p > 0.05). The correlations between age, duration, dosage, and OCT measurements are shown in Table 3.

Figure 3 The relationship between the duration of pregabalin use and the retinal layers.

Table 3 Correlation between OCT measurements and age, dosage, duration of use.

		RNLF
inferior	RNLF
superior	RNLF
nasal	RNLF
temporal	GCL-IPL
superior	GCL-IPL
inferior	GCL-IPL infero
nasal	GCL-IPL
Infero
temporal	GCL-IPL supero
temporal	GCL-IPL supero
nasal	GCC
superior	GCC supero
nasal	GCC
Infero
nasal	GCC
inferior	GCC
infero
temporal	GCC
supero
temporal	
Age	Correlation
coefficient	0.011	−0.431	0.119	−0.114	−0.431	−0.139	−0.309	−0.282	−0.386	−0.300	−0.210	−0.332	−0.348	−0.242	−0.160	−0.314	
p value	0.950	0.011	0.504	0.519	0.011	0.434	0.075	0.106	0.024	0.020	0.234	0.055	0.044	0.168	0.366	0.070	
Dosage	Correlation
coefficient	−0.061	−0.003	−0.122	−0.233	−0.195	−0.107	−0.187	-.020	−0.024	−0.151	−0.139	−0.186	−0.246	−0.149	−0.030	−0.085	
p value	0.731	0.985	0.491	0.184	0.268	0.549	0.290	0.912	0.893	0.396	0.435	0.293	0.161	0.399	0.866	0.631	
Duration of use	Correlation
coefficient	0.123	−0.048	−0.009	0.221	0.157	0.370	0.282	0.376	0.252	0.157	0.300	0.127	0.354	0.402	0.218	0.269	
p value	0.489	0.788	0.962	0.209	0.376	0.031	0.106	0.029	0.151	0.375	0.085	0.474	0.040	0.018	0.216	0.124	
Notes:

Spearman’s correlation coefficient.

RNLF, retinal nerve fiber; GCL-IPL, ganglion cell layer inner plexiform layer; GCC, ganglion cell complex.

Discussion

This study is the first to evaluate OCT parameters in patients with PGUD. We found a positive correlation between the duration of pregabalin use and the thickness of certain retinal layers (GCL-IPL inferotemporal, inferior; GCC inferonasal, and inferior). However, there were no significant differences between with PGUD and controls regarding the thickness of retinal layers.

Nonetheless, the association between pregabalin use at therapeutic doses and retinal layers has been previously investigated in fibromyalgia (FM) patients (Biçer et al., 2022). In that study, significantly lower RNFL thickness was detected in pregabalin users, and RNFL thickness was negatively correlated with the duration of pregabalin use. The authors recommended not to use this drug in FM patients with conditions that cause RNFL damage, such as diabetic retinopathy and glaucoma. There are differences between our study and the previous study that should be emphasized. The previous study was conducted in FM patients, comparing pregabalin users to non-users. In that study, the duration of the disease in the control group was not mentioned. Additionally, compared to our study, the mean age of the participants was higher, and the duration of pregabalin use and the dosage were lower. All these variables may affect retinal measurements.

The results of animal studies investigating the effect of pregabalin on the brain are contradictory. Several studies have demonstrated pregabalin’s anti-inflammatory, anti-apoptotic, and neuroprotective effects (Ha et al., 2008; Aşcı et al., 2016). The anti-apoptotic action of the drug is thought to occur via reducing caspase 3, p53, MAPK, and astrocyte proliferation, while the anti-inflammatory effect is believed to protect against oxidative damage through GABAergic modulation (Ha et al., 2008; Taha et al., 2020). However, other studies have reported that the drug is neurotoxic. Long-term, high dose pregabalin use in rats led to neuronal apoptosis and elevated oxidative stress in the cerebral cortex (Taha et al., 2020). In addition, two studies revealed that pregabalin increased gliosis in the frontal cortex (Elsukary et al., 2022; Sayin & Şimşek, 2018). This current study revealed that the duration of drug use is positively correlated with some retinal layer thickness. The results of previous studies are inconsistent, which may be attributed to the drug’s multi-acting nature, its effects on multiple neurotransmitters, the dose-dependent nature of certain effects like euphoria, and its unclear mechanism of action (Bockbrader et al., 2010).

We found that the majority of the OCT measurements were negatively correlated with age, but only RNFL superior, GCL-IPL superior, GCL-IPL superotemporal, GCL-IPL superonasal, and GCC inferonasal measurements were at a significant level. Previous studies have shown a significant decrease in GCL, IPL, and RNFL thickness with age (Chauhan et al., 2020). Decreases of 3% in GCL and 1–2 µm in RNFL per decade have been reported (Almarcegui et al., 2010; Chauhan et al., 2020). Our findings are consistent with the literature.

A confounding factor in our study is the higher smoking rates in the PGUD group compared to the controls. To address this, a General Linear Model (GLM) was created. The impact of smoking and pregabalin use on all OCT measurements was assessed using GLMs. Overall, the models did not significantly explain the variance in any of the OCT measurements. In a study by Çakır et al. (2017) no difference was detected in RNFL thickness among smokers. Another study also found no correlation between the smoking index (number of cigarettes smoked per day × smoking duration in years) and RNFL or GCC thickness (Aboud et al., 2022). In contrast, Moschos et al. (2016) reported that GCC thickness decreased in heavy smokers of more than 25 years. Evaluating previous studies shows that even if smoking affects the retinal layers, this effect occurs as a result of long-term and intensive use. Although we know the smoking status of participants in our study, the participants’ smoking years and amounts are unknown. In our study, almost all the participants were young adults. Therefore, long-term cigarette uses such as 25 years would be unexpected. Additionally, the smoking rate among drug users is very high, making it very difficult to find a PGUD patient who did not use tobacco.

The cross-sectional design and sample size are limitations of our study. We had to complete our study with limited number of patients due to the low prevalence of pregabalin use disorder when other comorbidities were excluded. In this study, we chose to investigate patients with pregabalin use disorder rather than those with pregabalin abuse to detect possible degenerative outcomes, so the minimum duration of use was 12 months that also limited our sample size.

Conclusions

In conclusion, we found a positive correlation between the duration of pregabalin use and the thickness of certain retinal layers, suggesting a potential neuroprotective effect of the drug. It is important to note that neurodegenerative changes associated with substance use disorders generally occur over long-term use. Therefore, longitudinal prospective studies are needed to explore the dose-duration relationship in individuals with PGUD, as well as the effects of pregabalin use on the macula and optic nerve. The 12-month minimum duration of pregabalin use in our study may have been too short to detect retinal layer changes, which often take years to manifest. Additionally, our study only included data from the left eye. Future studies incorporating data from both eyes could improve statistical significance.

Supplemental Information

Supplemental Information 1 Dataset codebook.

Additional Information and Declarations

Competing Interests

Author Contributions

Human Ethics

Data Availability

The authors declare that they have no competing interests.

Osman Hasan Tahsin Kılıç conceived and designed the experiments, performed the experiments, analyzed the data, prepared figures and/or tables, authored or reviewed drafts of the article, and approved the final draft.

Zehra Nur Bayram conceived and designed the experiments, performed the experiments, prepared figures and/or tables, authored or reviewed drafts of the article, data collecting, and approved the final draft.

Pelin Kiyat performed the experiments, authored or reviewed drafts of the article, and approved the final draft.

Omer Karti performed the experiments, analyzed the data, authored or reviewed drafts of the article, and approved the final draft.

Arzu Aral conceived and designed the experiments, performed the experiments, authored or reviewed drafts of the article, and approved the final draft.

Nazlı Deniz Munis performed the experiments, authored or reviewed drafts of the article, data collecting, and approved the final draft.

Berfin Gurbet Mutlu performed the experiments, authored or reviewed drafts of the article, data collecting, and approved the final draft.

The following information was supplied relating to ethical approvals (i.e., approving body and any reference numbers):

The Ethics Committee of İzmir Democracy University Buca Seyfi Demirsoy Training and Research Hospital. İzmir, Türkiye (2023/178).

The following information was supplied regarding data availability:

The data is available at Zenodo: Kılıç, O. H. T. (2024). Examination of Optical Coherence Tomography Findings in Patients with Pregabalin Use Disorder [Data set]. Zenodo. https://doi.org/10.5281/zenodo.13628486.

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
