# Peer review of "Examination of optical coherence tomography findings in patients with pregabalin use disorder"

_PeerJ, doi:10.7717/peerj.18395_

## Round 0.1 · original submission · Major Revisions

This paper has not yet been sent out for peer review. The language/grammar in the paper must be substantially improved prior to peer review. Please address the many language issues in the paper. Once these issues are addressed, then the paper will be reassessed to determine if the language has been improved sufficiently such that it is appropriate to send the paper out for peer review.

---

## Round 0.2 · Major Revisions

There are a number of serious issues raised by the reviewers that will need to be addressed. Most concerning are the comments from reviewer #1 regarding details related to potential differences in the experimental and control groups and how confounding factors (especially smoking) were (or were not) accounted for in the analyses. Reviewer #2 also requests more detail on the inclusion criteria, as well as figures demonstrating the results.

Reviewer 1 ·

Basic reporting

No Comments

Experimental design

The experimental design of this study presents some critical concerns regarding the selection and comparability of the control group to the PGUD group, which could significantly impact the validity of the conclusions drawn. These issues need to be addressed to ensure that observed differences between the two groups can be attributed reliably to the effects of PGUD. Please see the detailed comments below:

In the manuscript, details on how the control group was selected are sparse (Line 84). For studies not employing randomization, it is crucial to establish the comparability of the control group rigorously. A more thorough explanation of the criteria and methods used for selecting control participants is necessary. Ideally, controls should match the PGUD group in terms of demographic and clinical characteristics that could influence the study outcomes.

It is essential to provide a more comprehensive set of baseline characteristics for both groups. This should include not only age and smoking status but also other factors such as socioeconomic status, comorbidities, medication use, and lifestyle factors. These characteristics will help readers assess the comparability of the groups and the generalizability of the study findings.

The significant difference in smoking rates between the PGUD group (94%) and the control group (34%) highlighted in the results section could introduce substantial confounding. Smoking has been associated with changes in neurological and vascular health, which might influence the study's outcomes independently of PGUD. Strategies such as stratification or multivariable adjustment should be considered to mitigate the impact of this substantial confounding factor.

The reported duration of PGUD use (average 62.26 months, SD 40.09) suggests substantial variability within the PGUD group. This variability might introduce bias, as the effects of PGUD on the measured outcomes could correlate differently across shorter versus longer durations of use. More detailed information on the distribution of duration of PGUD use within the group is needed. Additionally, considering subgroup analyses or regression modeling to explore how different durations of use affect the outcomes could provide deeper insights and strengthen the study's conclusions.

Validity of the findings

The validity of the findings is affect by the issues raised in the experimental design (See section II) for more details.

Reviewer 2 ·

Basic reporting

Discussion:
• I noticed that the reference (Elgazzar, Elseady & Hafez, 2021) cited in in the discussion has been retracted (Elgazzar, Elseady & Hafez, 2023). It is crucial to replace this retracted reference with a current and credible source to maintain the integrity of the manuscript.
References:
• The reference (Elgazzar, Elseady & Hafez, 2021) listed in the references section has been retracted (Elgazzar, Elseady & Hafez, 2023). Please remove or replace this retracted reference with a credible and current source to maintain the accuracy of the manuscript.
Results:
• There are some sentences that are reported without any statistical values (lines 116, 125, and 133). The authors should know that any report provided in the study must be accompanied by statistical outcomes.
• If the mean of spherical equivalent refractive errors along with their standard deviations are available for both groups, please include them in result section to provide a more comprehensive comparison between the groups.

Experimental design

Methods:
• Any article that discusses imaging and thickness parameters must provide clinical images of the comparisons made to give more credence to the study. A montage image of the retinal layer thicknesses of subjects and controls can be provided.
• The manuscript should clarify whether both eyes or only one eye of each participant were included in the study. It is essential for understanding the study design and ensuring the validity and reliability of the results.
• The term "long-term" used in the manuscript is quite broad. Please specify the minimum duration required for participants to be considered as long-term users, and include this in the inclusion criteria.

Validity of the findings

No comment

Additional comments

This manuscript is well-written and provides valuable information that will benefit future studies and addresses a significant and underexplored area in the field of optical imaging and substance use disorders.
Overall, the manuscript has potential. After addressing the major revisions and concerns mentioned above, I recommend accepting the article for publication.

---

## Round 0.3 · Minor Revisions

Thank you for your previous responses to the reviewer comments. Please add the information in Table 1 requested in the latest review.

Reviewer 1 ·

Basic reporting

In Table 1, please list the numbers and percentages of employment and health insurance for the control group, for example, 0(0%), 35(100%). Please add not applicable(N/A) for dosage and duration of use.

Experimental design

The authors have made significant strides in addressing the concerns raised in the previous review.

Validity of the findings

The authors have made significant strides in addressing the concerns raised in the previous review.

---

## Round 0.4 · accepted · Accept

Thank you for your responses to the reviewer comments.